# Longitudinal deep sequencing informs vector selection and future deployment strategies for transmissible vaccines

**Megan E. Griffiths**[1,2]*, **Alice Broos**[1,2], **Laura M. Bergner**[1,2], **Diana K. Meza**[1,2], **Nicolas M. Suarez**[1], **Ana da Silva Filipe**[1], **Carlos Tello**[3,4], **Daniel J. Becker**[5], **Daniel G. Streicker**[1,2]

**1** MRC–University of Glasgow Centre for Virus Research, Glasgow, United Kingdom, **2** Institute of Biodiversity, Animal Health and Comparative Medicine, University of Glasgow, Glasgow, United Kingdom, **3** Association for the Conservation and Development of Natural Resources, Lima, Peru, **4** Yunkawasi, Lima, Peru, **5** Department of Biology, University of Oklahoma, Norman, Oklahoma, United States of America

\* m.griffiths.1@research.gla.ac.uk

## Abstract

Vaccination is a powerful tool in combating infectious diseases of humans and companion animals. In most wildlife, including reservoirs of emerging human diseases, achieving sufficient vaccine coverage to mitigate disease burdens remains logistically unattainable. Virally vectored "transmissible" vaccines that deliberately spread among hosts are a potentially transformative, but still theoretical, solution to the challenge of immunising inaccessible wildlife. Progress towards real-world application is frustrated by the absence of frameworks to guide vector selection and vaccine deployment prior to major in vitro and in vivo investments in vaccine engineering and testing. Here, we performed deep sequencing on field-collected samples of *Desmodus rotundus* betaherpesvirus (DrBHV), a candidate vector for a transmissible vaccine targeting vampire bat–transmitted rabies. We discovered 11 strains of DrBHV that varied in prevalence and geographic distribution across Peru. The phylogeographic structure of DrBHV strains was predictable from both host genetics and landscape topology, informing long-term DrBHV-vectored vaccine deployment strategies and identifying geographic areas for field trials where vaccine spread would be naturally contained. Multistrain infections were observed in 79% of infected bats. Resampling of marked individuals over 4 years showed within-host persistence kinetics characteristic of latency and reactivation, properties that might boost individual immunity and lead to sporadic vaccine transmission over the lifetime of the host. Further, strain acquisitions by already infected individuals implied that preexisting immunity and strain competition are unlikely to inhibit vaccine spread. Our results support the development of a transmissible vaccine targeting a major source of human and animal rabies in Latin America and show how genomics can enlighten vector selection and deployment strategies for transmissible vaccines.

**Data Availability Statement:** All aligned sequencing read files are available from the Sequence Read Archive (bioproject ID:

PRJNA732673). Sample data is available from Figshare (https://doi.org/10.6084/m9.figshare.15067884.v1). All other relevant data are within the paper and its Supporting Information files.

**Funding:** M.E.G. was supported by a Medical Research Council scholarship via the MRC-CVR PhD programme (MC_UU_12014/12) (https://mrc.ukri.org/). D.K.M. was supported by the Human Frontier Science Program (RGP0013/2018) (https://www.hfsp.org/) and the Mexican National Council for Science and Technology (CONACYT, 334795/472296) (https://www.conacyt.mx/). A.d.S.F. and N.S. were supported by the Medical Research Council (MC_UU_12014/12; MC_UU_12014/3). D.J.B. was supported by a National Science Foundation Graduate Research Fellowship, (NSF DEB-1601052) (https://www.nsf.gov/), the ARCS Foundation (https://www.arcsfoundation.org/national-homepage), and the Explorer's Club (https://www.explorers.org/). D.G.S., A.B. and L.M.B. were supported by a Wellcome Trust Senior Research Fellowship (217221/Z/19/Z) (https://wellcome.org/). The funders had no role in study design, data collection and analysis, decision to publish, or preparation of the manuscript.

**Competing interests:** The authors have declared that no competing interests exist.

**Abbreviations:** AAC, Apurímac; Ayacucho, and Cusco; AICc, corrected Akaike information criterium; DrBHV, *Desmodus rotundus* betaherpesvirus; GLMM, generalised linear mixed model; HCMV, human cytomegalovirus; kbp, kilobase pair; LCD, least-cost distance; MCMV, murine cytomegalovirus; MS, microsatellite; SNP, single nucleotide polymorphism; VTM, virus transport medium.

## Introduction

Zoonotic pathogens that transmit from wildlife to humans are challenging to both predict and control. Managing such pathogens within their wildlife reservoir hosts, rather than in dead-end hosts such as humans or livestock, has the potential to substantially reduce the associated health and economic burdens. Indeed, vaccination of wildlife has seen great successes, for example, in the control or elimination of rabies in carnivores using bait-delivered oral vaccines [1,2]. However, the ability to vaccinate sufficient proportions of wildlife populations to interrupt epidemiological dynamics remains a hurdle for most wildlife zoonoses. Vaccines that spread unaided between individuals, "transmissible vaccines," have been proposed as a theoretical solution [3–5]. Introductions of such vaccines into a small number of individuals (e.g., by bait or inoculation of captured animals) could yield higher coverage and spatial dissemination of immunological protection than traditional vaccines, potentially with minimal financial or operational investments in deployment.

Modern conceptions of transmissible vaccines favour using viral vectors to express an immunogenic protein of the target pathogen [4]. By deliberately exploiting naturally occurring, innocuous, and host-specific viruses as vectors, this approach circumvents risks of evolutionary reversion to pathogenic phenotypes, sometimes observed when attenuated virus vaccines evolve following unforeseen transmission [4,6]. However, the success of virally vectored transmissible vaccines is directly underpinned by the biological and epidemiological characteristics of the vector virus from which they are derived. A first challenge is avoiding cross-immunity between the vaccine vector and the naturally occurring ancestral virus (i.e., capacity for "superinfection") [3]. If multiple strains occur in nature and coinfections or serial coinfections (i.e., acquisition of a novel strain in an already infected individual) are readily observed, this would represent promising evidence for limited cross-immunity. A second challenge is that projecting how vaccines will spread within and between target populations is vital to determine whether they will attain sufficiently high coverage to dampen pathogen transmission. As considerable ethical and safety concerns preclude releasing engineered vaccines with undetermined dynamics of infection and spread, the prevalence and spatial distribution of wild-type ancestors provides the best possible approximation of candidate vector transmissibility [7]. Understanding these properties prior to the major investments required for vaccine construction and deployment is necessary, since patterns observed might reveal the need to consider alternative viral vector candidates.

Here, we studied the dynamics of *Desmodus rotundus* betaherpesvirus (DrBHV), a large DNA virus (Family: Herpesviridae) that infects common vampire bats [8,9]. Betaherpesviruses are promising transmissible vaccine vectors due to their host specificity, capacity to express foreign antigens, and largely innocuous effects on their hosts [3,4,10]. In most of Latin America, vampire bats are the primary reservoir of rabies virus, a zoonosis that causes sporadic and lethal outbreaks in humans and regular livestock mortality costing tens of millions of dollars annually [11]. This disease system is particularly suited to the use of transmissible vaccines, due to limited ability of existing interventions (bat culling and human/livestock vaccination) to reduce burden, the life history, and social behaviours of vampire bats that would facilitate vaccine transmission, and the epidemiology of rabies, which already exists on an extinction threshold [12,13].

While DrBHV shows promising levels of prevalence, individual vampire bats have multi-strain infections, necessitating investigation of strain-specific spatial structure and prevalence, as well as interactions between strains that might inhibit vaccine spread [8]. Here, we aimed to answer several unresolved questions that underpin the biological suitability of DrBHV as a viral vector, how it could be deployed to natural populations, and the extent that vaccine

spread would be predictable: (1) Do individual DrBHV strains reach sufficient prevalence and geographic range to plausibly dampen rabies transmission? (2) Are determinants of strain-specific prevalence consistent with weak competition among strains? (3) Is the geographic distribution of strains predictable by host genetic structure and expected landscape barriers to host movement? and (4) Do patterns of viral acquisition and viral genetic diversity within individual bats support superinfection and the latency and reactivation dynamics expected for betaherpesviruses? We answered these questions using a combination of longitudinal field studies of vampire bats from Peru, and deep sequencing.

## Results

### Variation in the prevalence and geographic range of DrBHV strains

Between 2013 and 2018, 132 saliva samples were collected from 110 vampire bats captured from 23 colonies (sample sites) in 8 departments of Peru (Fig 1A, inset). Unless otherwise noted, data from colonies within 10 km of each other (the expected vampire bat foraging distance [14]) were combined, creating 17 "groups" for later analysis (S1 Fig, S1 Table). A 12 kilobase pair (kbp) region of DrBHV, spanning the DNA polymerase (UL54), the glycoprotein B

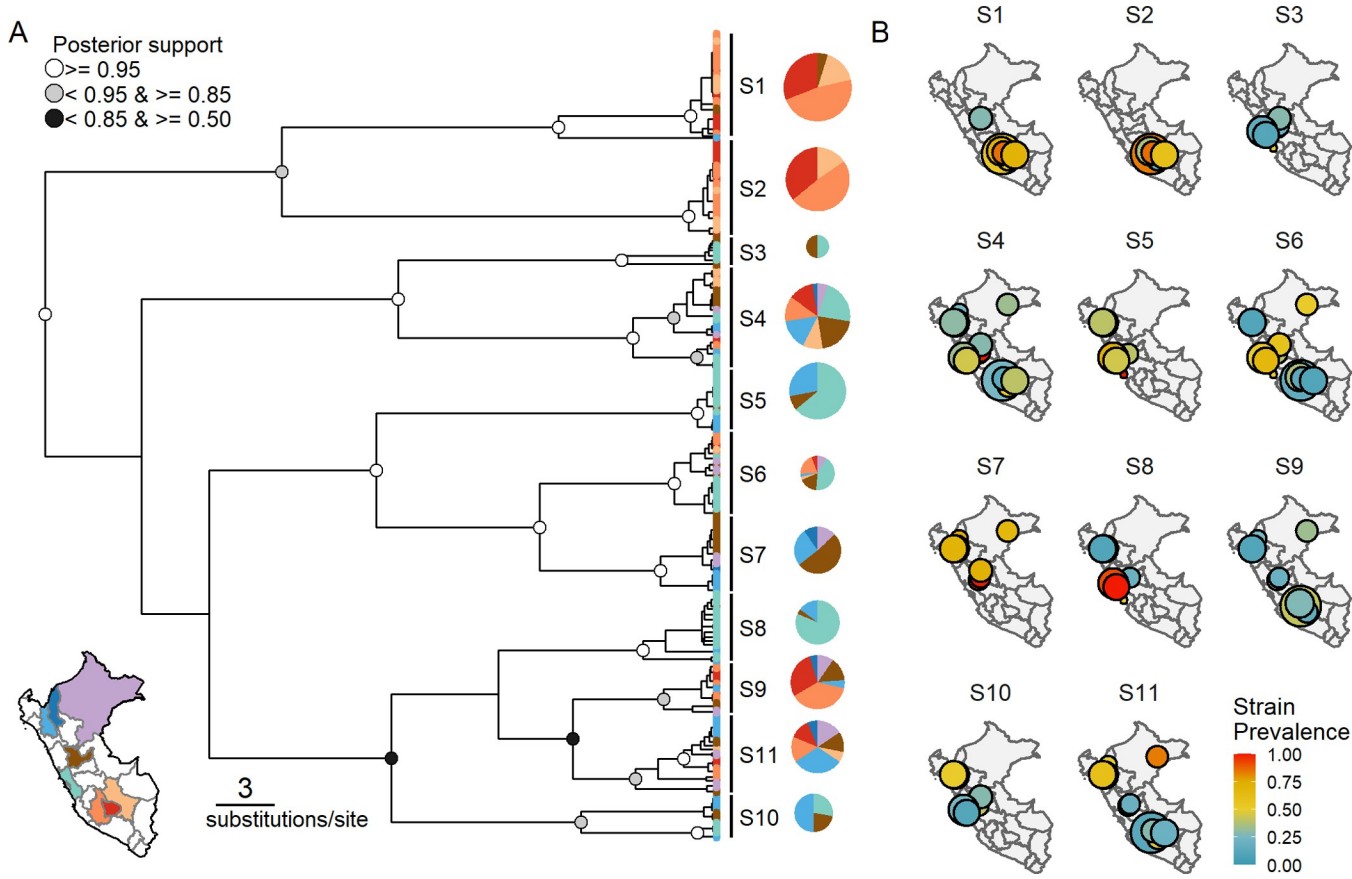

**Fig 1. Distinct geographic profiles of 11 circulating strains of DrBHV. (A)** Bayesian phylogeny of a 150-bp region of the UL55 glycoprotein B region of DrBHV. Tip colours and pie charts show the geographic origins of viruses within strains (see inset map). Pie chart size is proportional to the number of samples in which each strain was detected. Node colours represent posterior probability; support of <0.5 is not shown. **(B)** Prevalence maps of each DrBHV strain. Each map corresponds to a single strain, numbered S1 to S11 as in panel A. Points represents all colonies within 10 km with circle size proportional to sample size (range = 2 to 14, mean = 7.82) and colour indicating prevalence of infection (base map: https://gadm.org/maps/PER.html). Data underlying this figure can be found in (A) S1 Data and (B) S2 Data. DrBHV, *Desmodus rotundus* betaherpesvirus.

(UL55), and the terminase subunit (UL56), was amplified using a multiplex PCR in 7 over-lapping 2-kbp amplicons and sequenced in paired 250-bp reads on an Illumina MiSeq. Due to the ubiquity of multistrain infections, most consensus sequences assembled from these reads were chimeric. Therefore, analyses were either carried out at the read level, using Phyloscanner [15] to extract reads that fully spanned a 150-bp window, or using "poolseq" approaches, which do not require all reads to originate from the same genome (see below). Bayesian phylogenetic analysis of the most variable 150-bp window within UL55 revealed 11 DrBHV strains, which were further supported by affinity propagation clustering of the same 150-bp sequences [16,17] (Fig 1A, S2 Fig). Strains 1 and 2 shared a most recent common ancestor and were found almost exclusively in southern Peru in the departments of Apurímac, Ayacucho, and Cusco (AAC; Fig 1B). Remaining strains included those with a limited geographic range (e.g., strains 5 and 8, common within the coastal department of Lima [LMA] but absent or rare elsewhere) and those which were widespread across Peru (e.g., strains 4 and 6, found in every department sampled). One sequence did not cluster within the 11 defined strains but was most closely related to strain 1, although found in a different department to all other strain 1 samples. This was speculated to have arisen as the result of a recent recombination that has not yet become fixed in the population and was removed from subsequent analyses.

Averaged across the groups in which each strain was found, the observed prevalence of DrBHV strains ranged from 30% to 65%; however, local prevalence of some strains reached over 90% (Fig 1B). To understand the potential ecological and evolutionary determinants of this variation, we used a binomial generalised linear mixed model (GLMM) to investigate how prevalence was related to strain genetic diversity and geographic range, as well as the richness of locally cocirculating strains with which focal strains might compete. Both the relative nucleotide diversity and the geographic range of strains (measured as the minimum convex hull area of observed detections) were negatively associated with strain prevalence (Fig 2A and 2B, S2 Table). Comparison of test statistics to a null model derived by randomisation of the location meta-data associated with sequences showed that these relationships would not be expected to arise from chance (Fig 2). These results are consistent with the hypothesis that recently evolved strains (which all else equal would be expected to have lower genetic diversity [18]) attain high local prevalence and spread geographically, before being gradually replaced (S1 Text). In contrast, both at the coarse level of geographic group and the finer scale of sampling site, the number of locally cocirculating strains was unrelated to strain prevalence, implying that competition between strains is weak or undetectable (Fig 2C). We repeated this analysis using 12 strains defined by a different region of the genome, within UL54, to rule out any potential interference from selection acting on sequences from UL55. Nucleotide diversity remained negatively associated with prevalence ($\beta = -1.208$, SE = 0.559, z-value = $-2.162$, *p*-value = 0.031), but geographic range no longer showed any significant correlation. As above, the number of locally cocirculating strains had no significant relationship with strain prevalence, again supporting a lack of competition. These findings support the expectation that vaccine spread would be unimpeded by the circulation of preexisting DrBHV strains.

## Parallel population structure of DrBHV and vampire bat hosts

To understand the observed spatial distribution of DrBHV strains and the potential geographic barriers to vaccine spatial spread, we next approximated levels of viral gene flow between geographic groups of vampire bats using pairwise F statistics ($F_{ST}$, proportion of the total genetic variance contained in a subpopulation relative to the total genetic

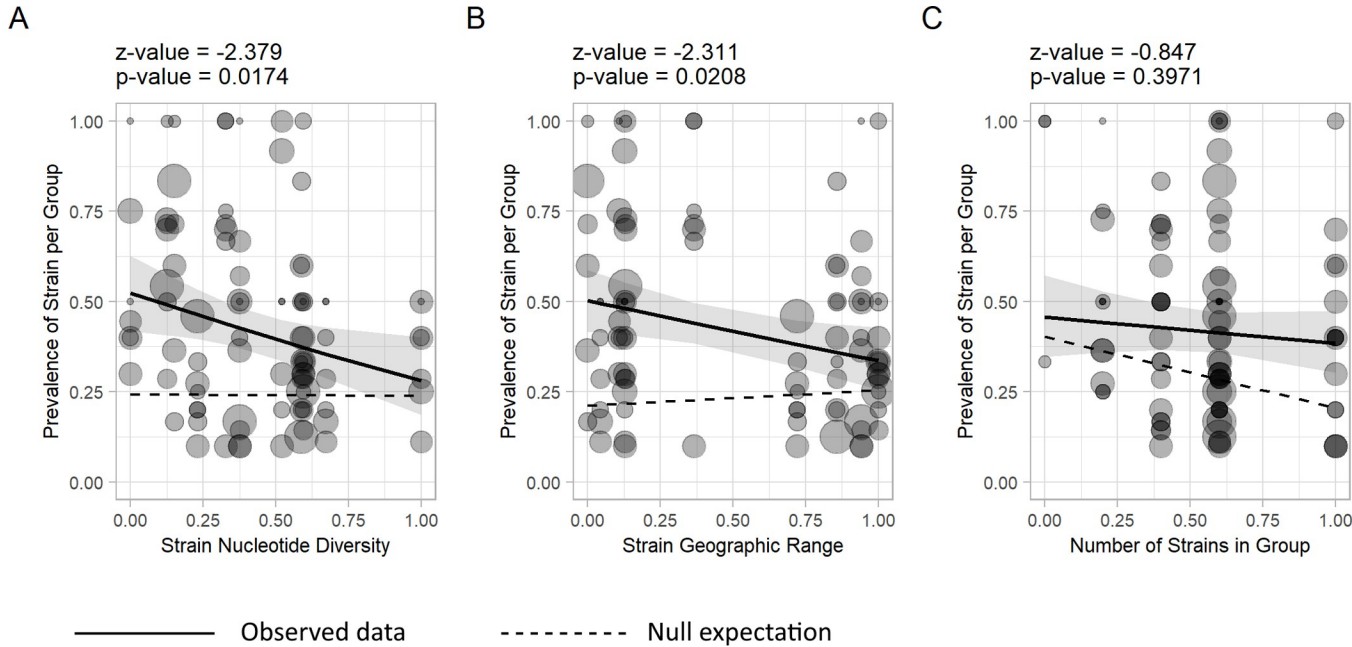

**Fig 2. Relationships between strain genetic diversity, geographic range, and prevalence suggest strain emergence, spatial diffusion, and replacement.** Predicted relationships between strain-specific prevalence and **(A)** strain nucleotide diversity, **(B)** geographic range, and **(C)** the number of strains present in each geographic group sampled (a measure of potential competition between strains). Lines and shaded areas represent the regression slope and 95% confidence intervals, respectively. Points are raw data, corresponding to the observed prevalence of each strain in each group where it occurred, with point size proportional to sample size (range = 2 to 24). The dashed lines on each graph show the null expectation, averaged over 1,000 tip randomisations. Data underlying this figure can be found in S2 Data.

variance). Our data comprised of short-read sequences with multiple DrBHV strains in each sample, which could not be reliably reconstructed into unmixed contigs. Therefore, each bat sample within a given geographic group was treated as a pool of virus samples and analysed using poolseq methodologies [19,20], and $F_{ST}$ calculated from these pooled single nucleotide polymorphism (SNP) read count data. Separate analyses considered SNPs found (a) across the full 12-kbp sequenced region; or (b) after removing reads from UL55 due to the increased possibility of selection on taking place on this gene encoding a viral glycoprotein. K-means clustering identified 3 groups, corresponding to (i) Huánuco, Amazonas, and Loreto; (ii) Lima and Cajamarca; and (iii) Ayacucho, Apurímac, and Cusco (Fig 3A). While low pairwise $F_{ST}$ within each of these 3 clusters shows internal mixing of strains, group (iii) was the most removed, suggesting greater genetic isolation of DrBHV in these southern populations. DrBHV $F_{ST}$ pairwise distance matrices (a) and (b) were strongly correlated (Mantel test: r = 0.83, $p$-value = $1 \times 10^{-4}$), suggesting that putative selection on UL55 did not alter the observed patterns of geographic relatedness. Overall, the population structure of DrBHV (a) mirrored that observed in vampire bat hosts using nuclear microsatellites (MS) (Fig 3, S1 Text; Mantel test on the pairwise distance matrices; r = 0.80, $p$-value = $1 \times 10^{-4}$; full Mantel tests on S3 Table) and in patterns of isolation predicted by a least-cost distance (LCD) model based on elevational barriers to host/virus gene flow (Fig 3, S1 Text; Mantel test on the pairwise distance matrices; r = 0.81, $p$-value = $1 \times 10^{-4}$). This parallel population structure is consistent with the expected host specificity of DrBHV and suggests that the geographic spread of a released vaccine strain would be predictable from and constrained by landscape barriers to bat dispersal [21].

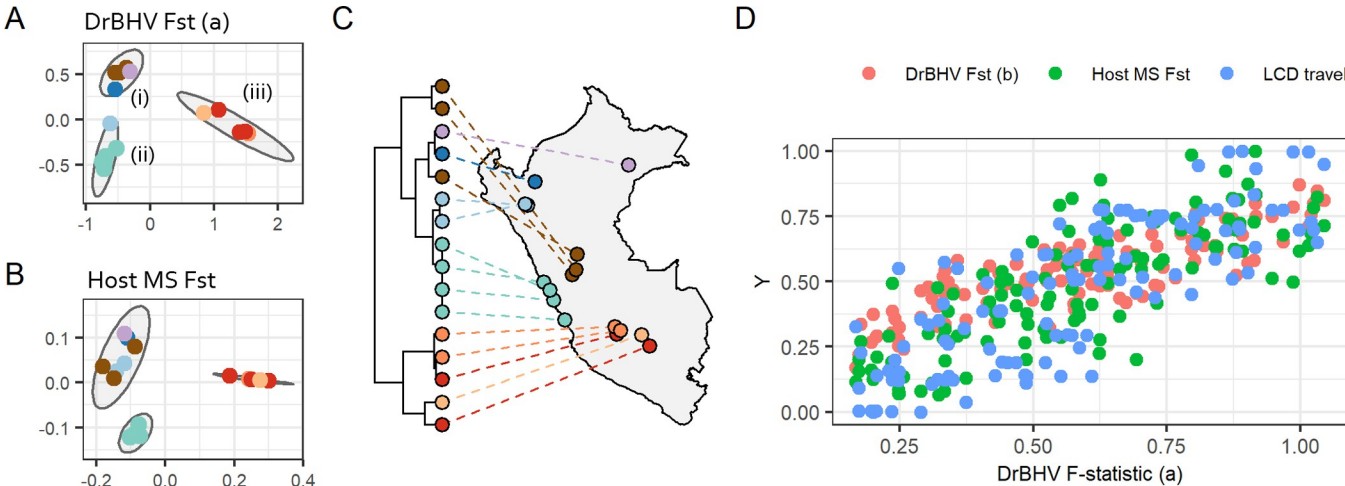

**Fig 3. Parallel population structure of DrBHV and vampire bat hosts.** Multidimensional scaling plots of the pairwise $F_{ST}$ between geographic groups of **(A)** DrBHV based on SNP read depth from UL54-56 and **(B)** *D. rotundus* MS data. Panel **(C)** shows a dendrogram representing LCDs between sampled colonies based upon elevation, with tips coloured by department of Peru (base map: https://www.evl.uic.edu/pape/data/WDB). Panel (A) shows 3 distinct clusters comprising (1) Loreto—purple, Amazonas—dark blue, and Huánuco—brown; (2) Lima—green and Cajamarca—blue; and (3) the southern region (Ayacucho, Cusco, and Apurímac). Panel (B) shows similar clustering with the exception that Cajamarca now clusters more closely with (1) than with (2). Panel **(D)** shows strong correlations between pairwise DrBHV Fst (a—based on the full complement of SNPs) and 3 other distances matrices: DrBHV Fst (b—SNPs from UL55 not included) (red), host microsatellite Fst (blue), and pairwise distances based on the least-cost landscape model (green). Data underlying this figure can be found in S3 Data. DrBHV, *Desmodus rotundus* betaherpesvirus; LCD, least-cost distance; MS, microsatellite; SNP, single nucleotide polymorphism.

## Longitudinal sampling of vampire bats shows strain acquisition, long-term persistence, and within-host diversification of DrBHV

Multistrain infections were widespread across demographic groups of vampire bats. Among the 132 DrBHV-positive vampire bat saliva samples, 79% ($n = 101$) were infected by multiple strains (average: 2.4 strains per individual; range: 1 to 5 strains per individual, Fig 4A). A Poisson GLMM showed no significant effects of sex or age on the number of strains detected ($p$-value > 0.05 for both; S4 Table). Rather, the number of strains detected within individuals was positively correlated with the total number of locally available strains ($\beta = 0.098$, SE = 0.048, z-value = 2.055, $p$-value = 0.039).

We further investigated the dynamics of strain acquisition and loss using samples from 20 individually marked bats that were recaptured 1 to 2 times between 2013 and 2018. Deep sequencing revealed 12 instances of strain acquisition, occurring between 3 months and 4 years after initial sampling (Fig 4C). We used a Poisson GLM to test effects of bat sex, the time interval between sampling periods, the number of strains infecting a bat at its initial sampling (hereafter, "initial strain richness"), and the proportion of local strains remaining available for acquisition (i.e., those strains that were present in the area, but did not yet infect the focal bat) on how many strains were gained at later sampling instances. Bat sex, sampling interval, and initial strain richness had no effect on the number of strains gained by an individual. However, bats which had a small fraction of locally available strains at initial sampling acquired significantly more additional strains at later time points ($\beta = 8.463$, SE = 4.293, z-value = 1.971, $p$-value = 0.048; Fig 4B) than those with a higher fraction of locally available strains. Together with our earlier findings suggesting noncompetitive coexistence among strains at the population level (Fig 2), these results strongly suggest the absence of immunological barriers that would inhibit infection by novel strains, even in already multiply infected bats. A total of 11 out of 20 bats also apparently lost one or more DrBHV strains at later sampling instances, and

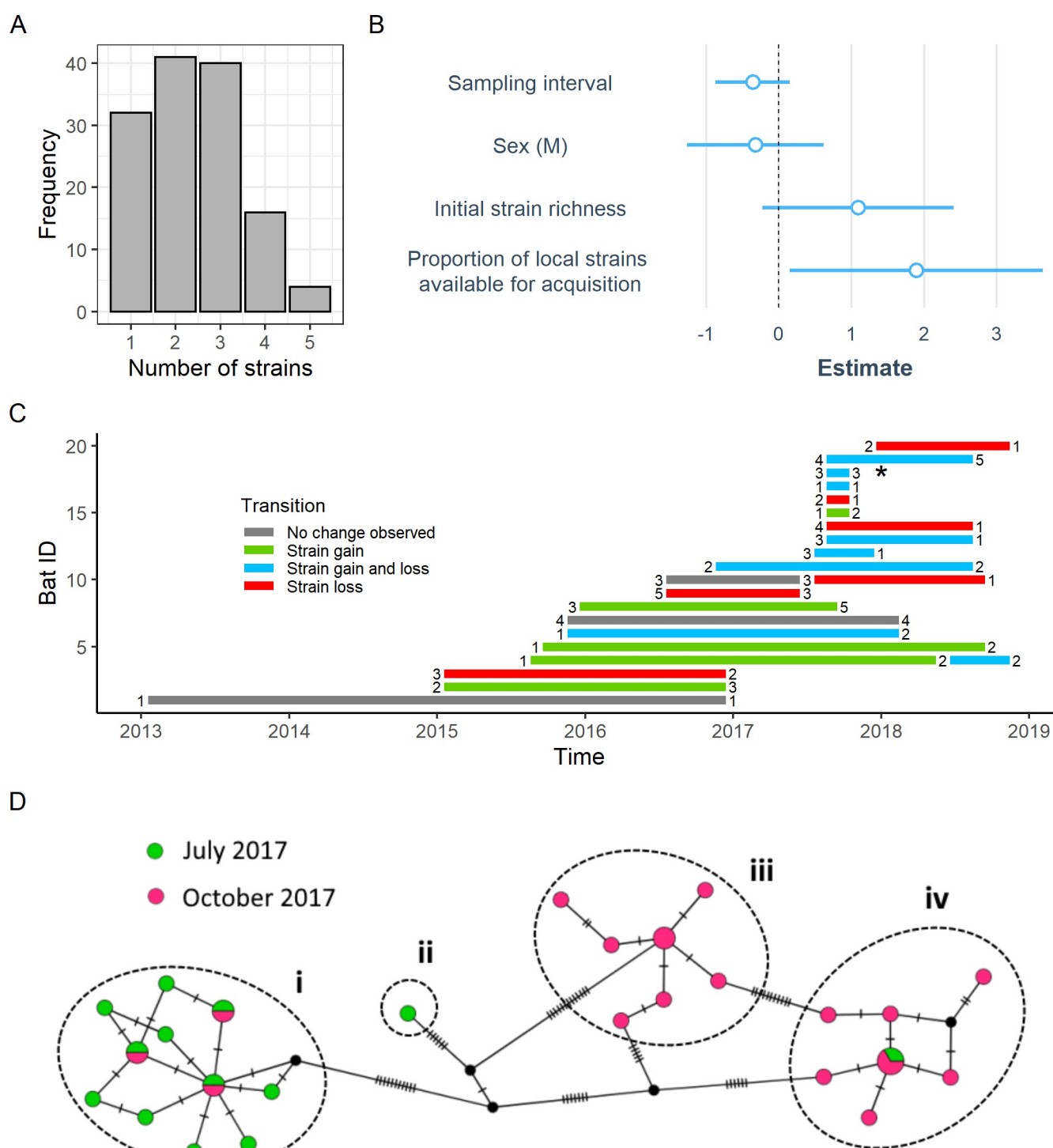

**Fig 4. Longitudinal sequencing shows strain acquisition and within-host persistence. (A)** Histogram of the number of DrBHV strains detected in each individual bat at a single time point. **(B)** GLM estimates of the effect of time between samples, bat sex, and strain availability on the number of strains gained between sample points shows that strain acquisition depends on local strain availability. **(C)** The total number of strains found in longitudinally sampled bats through time and whether strains were gained (green), lost (red), both (blue), or remained unchanged (grey) in the intervening period. **(D)** Haplotype network of strains found in Bat 9626 (marked with * in panel (C)), with separate strains circled. Tick marks represent the number of nucleotide changes between haplotypes. A single circle with both colours represents an identical variant found at both time points. The strain in (D-i) shows a decrease in intrastrain diversity between sampling points. Strain (D-ii) is found only at time point 1, demonstrating potential strain loss, while (D-iii) is only found at time point 2, illustrating strain acquisition. (D-iv) shows an increase in intrastrain diversity over time, indicating within-host persistence and evolution. Data underlying this figure can be found in S4 and S5 Data. DrBHV, *Desmodus rotundus* betaherpesvirus; GLM, generalised linear model.

none of the factors tested above explained the probability of strain loss or the number of strains lost.

Given that lifelong, persistent infection of salivary glands with varying levels of replication is a common feature of human and murine cytomegalovirus (HCMV and MCMV) [22–25], we further investigated the dynamics of within-host persistence of DrBHV. In 16/20 longitudinally sampled bats, the same strain was detected across sampling instances and up to 4 years post initial detection (the maximum duration permitted by our sampling design). Since the frequency or efficiency of strain replication would be expected to promote evolutionary diversification, we next studied the genetic diversity of individual strains within-hosts (i.e., intrastrain diversity) through time [26]. Nine paired samples with persistent infections showed either increases or decreases in intrastrain diversity over time, suggesting viral replication and genetic bottlenecks potentially related to the establishment of latency, respectively. For example, Bat 9626 (Fig 4D) exemplified both patterns of intrastrain diversity (increases: 4D-iv; decreases: 4D-i) as well as strain gain (4D-iii) and strain loss (4D-ii) over a 3-month time period. Multiyear infections with evolutionary diversification but variable intrastrain genetic diversity within sampling instances is broadly consistent with long-term viral persistence within individual hosts via periodic phases of latency and reactivation.

## Discussion

Focusing on a betaherpesvirus of common vampire bats, we demonstrate the power of spatially replicated, longitudinal deep sequencing to reveal cryptic strain-specific patterns in prevalence and geographic range while providing direct evidence for the within-host processes governing viral maintenance over long time periods. Viewed in light of DrBHV as a proposed viral vector of a transmissible rabies vaccine, our results show that vaccine spread between bats could achieve high population-level coverage and is unlikely to be inhibited by preexisting immunity or competition with naturally circulating strains. Our geographic and population genetic analyses also reveal how geographic heterogeneity in strain presence and spatial connectivity may be exploited to guide vaccine deployment. Such a vaccine, if developed, could be transformative for the prevention of human and animal rabies in Latin America.

A major concern with using naturally occurring viruses as transmissible vaccine vectors has been that cross-immunity from wild-type relatives could impede vaccine spread [3]. In our study, widespread multistrain infections, acquisitions of novel strains in longitudinally sampled bats, and evidence that neither strain acquisition nor population-level prevalence appeared to be limited by cocirculating strains suggests an absence of cross-immunity among DrBHV strains. This conclusion implies that the very high prevalence of natural DrBHV infection is unlikely to impede the spread of a hypothetical DrBHV-vectored rabies vaccine within vampire bat populations. The strain-specific geographic structure we observed could further enable selection of vector strains that are rare or absent in a region targeted for vaccination, thereby reducing the already unlikely potential for strain competition, and facilitating genomic monitoring of vaccine spread. A remaining uncertainty, however, is that natural selection is expected to purge antigenic inserts from viral vectors. Although techniques to reduce the loss of inserts are advancing [27], this phenomenon has potential to create competition between circulating vaccine strains and "ex-vaccine" strains that have reverted to wild type and may have a fitness advantage over remaining vaccine strains [28].

Although capacity for superinfection is advantageous for the spread of transmissible vaccine vectors, multistrain infections open possibilities for recombination, which has been cited as a safety concern [7]. Although we were unable to formally evaluate recombination in our data set due to the sequencing methods employed, recombination is common in herpesviruses

[7,29,30] and may explain a divergent sequence that was observed only in a single individual. Thus, while recombination in DrBHV would be unsurprising, we are unaware of instances where recombination among betaherpesviruses has changed host range or pathogenicity. Indeed, it is expected that strong host specificity precludes exchange of genetic material along betaherpesviruses adapted to different host species since they are unlikely to co-occur in the same host. Regardless, we encourage in vitro tests of vaccine host range across a range of cell types derived from different host species and modifications of vaccine design if necessary.

Our results also provide insights into the expected dynamics of spatial spread that might follow release of a DrBHV-vectored rabies vaccine. Patterns of DrBHV strain structure mirrored the population genetic structure of vampire bats. One possible exception to this pattern was the northern Peruvian department of Cajamarca, which clustered with other northern sites according to bat MSs but instead with sites in the coastal department of Lima in DrBHV genetic structure (Fig 3). Importantly, however, a more comprehensive population genetic analysis showed host gene flow between Lima and Cajamarca, and our landscape model also supported connectivity between these regions; both findings are consistent with the sharing of DrBHV genetic diversity among these regions [21]. These commonalities reinforce the previously suggested host-specificity of DrBHV and suggest that barriers to bat gene flow limit DrBHV spread [8]. Such barriers are likely to include mountain ranges that restrict bat movement, as shown by the correlation between both host and virus population structures with a LCD of travel model based on elevation. Moreover, the strong correlation between landscape topography and viral genetic structure implies that virus spread might be anticipated even in the absence of host genetic data, facilitating eventual large-scale vaccine deployments by identifying relatively isolated areas that would not otherwise be reached by unaided spread. In the short term, isolated areas such as the one we identified in southern Peru might be exploited for small-scale trials, since potential for vaccine escape would be diminished.

We also observed that high prevalence DrBHV strains tended to have low genetic diversity and small geographic ranges (Fig 2). One possible explanation for this pattern is that strains emerge, attain high prevalence, and spread geographically before gradually declining in prevalence, consistent with predicted early SILI dynamics [31], which herpesviruses are thought to follow. Since we were unable to estimate strain age directly, a key assumption underlying this interpretation is that strain nucleotide diversity increases through time. Although this relationship is expected from theory, selection and population size may also affect nucleotide diversity [18]. While we cannot completely rule out these explanations, if nucleotide diversity was predominantly determined by current viral population sizes, we would expect a positive relationship between strain nucleotide diversity and prevalence rather than the negative relationship we observed. While selective sweeps that reduce genetic diversity might obscure the increase of nucleotide diversity through time, the relationship that we observed between diversity and prevalence was robust and was also observed when using strains defined using a section of UL54, which should experience less selective pressure than the UL55-enocded glycoprotein B. Encouragingly, a similar pattern between strain age and prevalence has been observed for HCMV, where genotype gB4 is consistently the least prevalent genotype [32–36], despite being estimated as the oldest of the 4 glycoprotein B genotypes (S3 Fig, S1 Text). Although we emphasise the preliminary nature of this conclusion, if DrBHV-vectored vaccines behave similarly to wild-type DrBHV strains, our results suggest they would rise in prevalence, spread predictably within the bounds of bat population structure, and eventually become minor variants.

Our study confirms the expectation from other betaherpesviruses [37,38] that DrBHV establishes persistent infections within individuals. Additionally, given that intrastrain diversity appears likely to correlate with viral load (S1 Text) and shedding as might be expected [39], our results suggest that strain-specific virus shedding varies substantially over relatively

short time periods, even during "active" infection. Unlike HCMV and MCMV, which cause lifelong infections, we observed several putative instances of strain loss. It is unclear whether these reflect strains which were present at undetectable levels at the time of sampling (i.e., latent in other bodily compartments), were outcompeted in PCR amplification by other strains undergoing higher replication, or true strain loss from the host. However, even if DrBHV infection proves not to be "lifelong," the multiyear, latency, and reactivation dynamics suggested by our data imply that a DrBHV-vectored vaccine might enable prolonged protection against rabies. While vampire bats can live up to 17 years in the wild, typical lifespans are shorter (<5 years) [40], meaning that vaccines would in principle periodically boost during DrBHV reactivations to protect vampire bats against rabies through most of their lives, especially considering some persistence of rabies immunity beyond putative clearance of the DrBHV vaccine or loss of the antigenic insert. Such boosting could reduce the frequency at which long-term redeployment of vaccine would be required, as generations of bats should remain immune throughout their lifetime.

An unavoidable limitation of our strain-level analyses was that they required grouping viruses into epidemiologically relevant and tractable strains. In the absence of antigenic data, we used a clustering algorithm to operationally and reproducibly define strains as genetically similar clusters, and these were strongly supported in our phylogenetic analysis. Nevertheless, it is likely that some of the patterns we observed could be sensitive to how strains are defined. In particular, increasingly narrow strain definitions would be expected to create smaller observed geographic ranges, and additional sequencing might lead to discovery of new strains or separation of existing strains that could also influence their inferred geographic range. Importantly, however, altering strain definitions would not be expected alter our conclusions on the geographic spread of DrBHV (which did not require strain definition), superinfection, or within-host dynamics.

Several additional lines of investigation are needed to further develop DrBHV as a transmissible rabies vaccine. First, we emphasise that our study explored patterns in wild-type DrBHV strains to derive expectations for the dynamics of a still-hypothetical vaccine. Some of our findings, such as patterns of spatial structure, would not be expected to differ between wild-type and vaccine strains since they depend more on bat dispersal behaviour than viral genotype. Other findings, such as the capacity for superinfection and the duration of infections within hosts, may more plausibly be affected by genetic manipulation and should be retested after isolating DrBHV and engineering it with a view to maximise immunogenicity and evolutionarily stability. Fortunately, the established use of BHVs as vaccine vectors provides an informed starting point for this process [10,41–43]. Second, the ubiquitous presence of DrBHV in apparently healthy vampire bats, the survival of bats with long-term infections, and the fact that most BHVs cause minimal disease in healthy hosts support the conclusion that DrBHV is innocuous. Nevertheless, pathological studies are necessary to rule out sublethal fitness costs, which may have implications for animal welfare. Third, while our data show that the spatial spread of a DrBHV vaccine may be predictable, the timescale of transmission and spatial spread remain critical unknowns. These might be addressed via a combination of captive bat experiments and field experiments using genetically tagged, but otherwise unmanipulated, DrBHV strains [44]. Potential interactions between vaccine and ex-vaccine strains and their implications for the long-term dynamics of vaccine spread and deployment strategies will also require experiments with captive bats and mathematical models, respectively. In the interim, models exploring strategies for transmissible vaccine deployment should incorporate sensitivity to the possible consequences of antigen loss. Finally, our sequencing focused on the glycoprotein B region of DrBHV. Long-read sequencing could provide deeper knowledge of strain diversity and distribution while providing longer sequences for phylogenetic analyses.

Once dynamics of vaccine infection and spread are understood, pilot field studies and models will be needed to inform further large-scale deployment.

More broadly, viruses that multiply infect individual hosts are widespread in nature, including important human, livestock, and wildlife pathogens such as dengue viruses [45], human herpesviruses [46,47], bovine and equine rotaviruses [48,49], and avian influenza viruses [50,51]. Our study indicates the importance of differentiating multiple strains when evaluating virus dynamics. For example, PCR of a conserved region of DrBHV followed by Sanger sequencing suggested a countrywide prevalence of 97% [8], far higher than the strain averages observed here. Grouping multiple strains also would have vastly overestimated geographic range and disguised within-host dynamics, since all bats would have appeared to be persistently infected with one strain [8], as opposed to a complex mix of persistence, strain gain, and potentially strain loss over time [31]. Further, estimates of genetic structure from chimeric sequences derived from multistrain infections would have been uninformative, limiting the use of consensus sequences to infer viral spread. Herpesviruses in particular are often used as model systems for studying viral transmission dynamics in wild populations or for studying host connectivity [52–54]. If multistrain infections are as ubiquitous in other host species as we observed in vampire bats, serological, PCR, or even Sanger sequencing (without cloning) would be unlikely to provide a realistic picture of virus population dynamics.

This study demonstrates the use of genomics as a crucial early step to evaluate the suitability of transmissible vaccine vectors for wildlife diseases prior to vaccine development. Our findings offer new lines of evidence that support the feasibility of a DrBHV-vectored rabies vaccine and reveal opportunities for spatially informed vaccine deployment to wild bat populations. Considering the potential for transmissible vaccines to inexpensively mitigate the large and recurring human health and agricultural losses caused by vampire bat rabies across Latin America, further investment in the development and testing of vaccine vectors would seem warranted.

## Methods

### Sample collection

Vampire bats were sampled between 2013 and 2018 at 23 colonies (sample sites) across 8 departments (Amazonas, Apurímac, Ayacucho, Cajamarca, Cusco, Huánuco, Lima, and Loreto) of Peru. Bats were captured using mist nets and harp traps and then placed in individual cloth bags before processing and sampling [55]. Bats were aged as juvenile, subadult, or adult by observation of epiphyseal–diaphyseal fusion [56]; sex and reproductive status were also recorded. Saliva swab samples were collected by allowing bats to chew on sterile cotton-tipped wooden swabs (Fisherbrand, UK) for 10 seconds. Whole blood samples were also collected on swabs after puncturing the propatagial vein with a sterile 23-gauge needle. All swabs were stored in either 1-ml RNALater (Ambion, UK, 12 hours at 4˚C and then moved to −80˚C), virus transport medium (VTM, phosphate-buffered saline supplemented with 10% fetal calf serum, and double-strength antibiotic/antimycotic [200 U/mL penicillin, 200 g/mL streptomycin, and 0.5 g/mL fungizone amphotericin B]), or PBS and then stored in dry ice in the field and at −80˚C in the lab, until further analyses. Capture and sampling of bats was approved by the Research Ethics Committee of the University of Glasgow School of Medical, Veterinary and Life Sciences (Ref081/15) and by the University of Georgia Animal Care and Use Committee (A2014 04-016-Y3-A5). Field collections were authorised by the Peruvian government (RD-009-2015-SERFOR-DGGSPFFS, RD-264-2015-SERFOR-DGGSPFFS, RD-142-2015-SERFOR-DGGSPFFS, and RD-054-2016-SERFOR-DGGSPFFS).

## Nucleic acid extraction

Nucleic acid extractions from swabs were performed on a Kingfisher Flex 96 automated extraction instrument (Thermo Fisher Scientific, UK) with the BioSprint One-For-All Vet Kit (QIAGEN, UK) using a modified version of the manufacturer's protocol for purifying viral nucleic acids from swabs [55].

## Primer design and screening of samples

A 12-kbp region of the DrBHV genome, spanning the putative UL54-56 coding regions, was selected for multiplex amplification due to the high variability of UL55, which encodes glycoprotein B and is often used in HCMV strain typing [36]. Primers were designed using Primal scheme [57] on the consensus sequence from a DrBHV short-read sequencing run spanning the full genome [8] (SRA SRR11789720) to produce 7 pairs of primers with overlapping 2-kbp amplicons (with an overlap of approximately 200 bp) spanning the chosen region. The primers were divided into 2 sets (odd and even numbered) for amplification reactions (S5 Table). PCRs were carried out using a high fidelity Q5 polymerase and 5× reaction buffer with 2.5μL of extracted DNA per reaction. The PCR programme consisted of a 30-second 98˚C initial denaturation, followed by 45 cycles of denaturation at 98˚C (15 seconds) and 65˚C (5 minutes) annealing and extension and concluding with a 15-minute final extension at 65˚C. PCR products were tested for amplification on a 1% agarose gel for 2-kb bands.

A total of 132 saliva samples from a range of colonies and years were positive for both sets of primers and were selected for sequencing. These samples included 42 saliva samples from 20 longitudinally sampled bats (2 to 3 samples per bat; 3- to 40-month intersampling period) for our studies of the temporal dynamics of within-host strain diversity.

## Library preparation, sequencing, and sequence assembly

PCR products from the same sample were combined and purified using the QIAGEN PCR purification kit (QIAGEN), and the DNA concentration measured by Qubit (Thermo Fisher Scientific). Purified PCR products were adjusted to 100 ng in 55 μL for library preparation. The 2-kb amplicons were sheared in size for optimal Illumina sequencing, using a Covaris S220 Sonicator at peak power 450 watts, Duty factor 11.5, and 1,000 cycles per burst for 50 seconds. Peak size was 380 bp using an Agilent (UK) 2200 Tapestation system and within the optimum size range for paired 250-bp reads to be produced. Sequencing libraries were prepared using the KAPA HTP/LTP Library Preparation Kit (Roche, Germany) with NEBNext Multiplex Oligos for Illumina (Dual Index Primers Set 1) from NEB, UK, and sequenced on the Illumina MiSeq System (US) to produce an average of approximately 185,000 paired end reads of 250 bp per sample. Illumina reads were filtered, cleaned, and trimmed using *trim_galore* (https://github.com/rjorton/Allmond) [58,59] and aligned to the reference sequence (section of the consensus sequence from previous metagenomic sequencing [8]) using *bowtie2* [60] and *samtools* [61], producing a bam alignment for each of the samples.

## Delineation of circulating strains

Phyloscanner [15] was used to extract sequences from the bam alignments of each sample in 150-bp windows. This window size was the largest possible from a read length of 250 bp. Scanning across the amplified 12-kbp region identified an area of highest variability from 7,900 to 8,050, within the putative UL55 region. Phyloscanner was run on this window with a minimum read count of 250 reads and filtering of unpaired reads in order to discard low-quality sequences and those that may exist due to sequencing error. BEASTv1.10.4 [62] was used to

produce a Bayesian phylogeny of these sequences using a HKY+gamma substitution model as selected by Jmodeltest2 [63] using the corrected Akaike information criterium (AICc). Tree prior Coalescent: Bayesian skyline, and strict clock was used for the final $1 \times 10^{-7}$ generations run with a sampling frequency of 1,000 and a burn-in of 10%. These clock and tree priors were selected after testing both tree prior constant and Bayesian skyline, with strict clock at $1 \times 10^{-6}$ generations, and finding both models to be in agreement, due to the lack of rigid assumptions about population and evolutionary dynamics. Tip dates were not included due to the short duration of sample collection compared to the estimated rate of evolution of the virus, as well as sampling date being unlikely to represent date of infection, and insufficient information was available to inform internal node priors. As such, branch lengths represent substitutions per site and are not timescaled. The maximum clade credibility tree was visualised in R [64] using *ggtree* [65]. The 11 strains of DrBHV were defined using *apcluster* in R [66], using both the distance matrix extracted from the DrBHV phylogeny and the distance matrix from the sequence data alone. This method was also used to define strains based on a 150-bp section of UL54.

## Phylogeographic analysis

Allele read count data were extracted from each sample using the mpileup function of the SNP calling software "bcftools" [67]. SNPs with more than 2 alleles were removed, and the read counts pooled by origin group of each sample. The data were then analysed using *poolfstat* in R [19], which performs well even at low sample size and low coverage. Pairwise $F_{ST}$ between pooled groups was calculated based on (a) all SNPs present in the 12-kb sequenced region covering putative UL54-56 coding regions; and (b) SNPs found in only UL54 (DNA polymerase) and UL56 (terminase subunit), with those from UL55 (glycoprotein B) removed due to the high likelihood of selection influencing this gene. Pairwise $F_{ST}$ values were visualised in R using *ggplot2* [68].

## Statistical analyses

A GLMM with binomial distribution was used to determine the influence of DrBHV strain nucleotide diversity, strain range, and number of other strains in the same area on the prevalence of each strain in each geographic group. Nucleotide diversity was calculated for each strain using DnaSP [69] (S1 Text), and strain geographic range was calculated using *sp* in R [70], as the convex hull area based upon the coordinates of the sample sites in which each strain was found. Strain was used as a random effect. This analysis was repeated on strains defined as above for the most variable 150-bp region of UL54, in order to test the effects on prevalence without the potential influence of selection on UL55 (glycoprotein B). A randomisation test was also carried out, in which tree tips were randomly reassigned location metadata. The GLMM was applied to each random data set to produce a null distribution, which can be found in S2 Data.

The effects of bat age, sex, and the number of locally circulating strains on the number of strains within individual bats were investigated in a Poisson GLMM. Age was modelled as a categorical variable with 2 classes separating adults from younger bats (juveniles and subadults). Group was used as a random effect. All GLMMs were fit using the package *lme4* [71] and visualised using the package *ggeffects* in R [72].

For longitudinal samples, the time between sampling points, bat sex, initial strain richness, and the proportion of local strains available for acquisition were evaluated using a Poisson GLM [71] on the number of strains gained/lost between sampling points. The results of the model were visualised using *jtools* in R [73]. Haplotype networks of longitudinal samples were created using DnaSP [69] and PopART [74]. Bat age was not tested in this model, as all bats

that were longitudinally sampled were adults. Poisson GLMs were checked for overdispersion, and all models were checked for the normality of residuals.

## Supporting information

**S1 Fig. Vampire bat colonies within 10 km sorted into groups.** Map of Peru showing the grouping of sample colonies within 10 km of each other into 17 groups (base map: https://gadm.org/maps/PER.html).
(TIFF)

**S2 Fig. Affinity propagation clustering produces 11 clusters of DrBHV.** Plot of the 11 clusters produced by affinity propagation clustering and the similarity of each sequence (yellow = high similarity, dark blue = low similarity). Grey and red bars are used to separate clusters. The dendrogram shows relatedness between the clusters. Data underlying this figure can be found in S6 Data. DrBHV, *Desmodus rotundus* betaherpesvirus.
(TIFF)

**S3 Fig. Age determination of HCMV glycoprotein B strains.** Bayesian phylogeny of HCMV glycoprotein B sequences with the age of nodes (million years). Panine herpesvirus 2 used as an outgroup with a set divergence date of 3.8 mya based on the divergence date of host species. Node bars represent 95% highest posterior densities for date estimates. Data underlying this figure can be found in S7 Data. HCMV, human cytomegalovirus.
(TIFF)

**S1 Table Coordinates of sample sites (bat colonies) and corresponding group designations.**
(XLSX)

**S2 Table. Results of GLMMs testing the effects of nucleotide diversity, geographic area, and strain richness on strain prevalence.** GLMM, generalised linear mixed model.
(XLSX)

**S3 Table. Mantel tests comparing virus and host population structure and geographic determinants.** Comparisons between DrBHV pairwise $F_{ST}$ **(a)** and **(b)**, host MS pairwise $F_{ST}$, and LCD for travel between groups. DrBHV, *Desmodus rotundus* betaherpesvirus; LCD, least-cost distance; MS, microsatellite.
(XLSX)

**S4 Table. Statistical tests and results of within-host DrBHV analyses.** DrBHV, *Desmodus rotundus* betaherpesvirus.
(XLSX)

**S5 Table. Primers used for multiplex amplification of DrBHV.** DrBHV, *Desmodus rotundus* betaherpesvirus.
(XLSX)

**S1 Text. Supporting methods and results text.**
(DOCX)

**S1 Data. Tree file for the DrBHV phylogeny shown in Fig 1.** DrBHV, *Desmodus rotundus* betaherpesvirus.
(TRE)

**S2 Data. Prevalence of each strain of DrBHV per geographic group and results of null model statistical tests.** DrBHV, *Desmodus rotundus* betaherpesvirus.
(XLSX)

**S3 Data. Distance matrices used to create Fig 3.**
(XLSX)

**S4 Data. Longitudinal samples strain data used to create Fig 4B and 4C.**
(CSV)

**S5 Data. Haplotype file for longitudinal samples taken from bat 9626 used to create Fig 4D.**
(NEX)

**S6 Data. Distance matrix based on DrBHV 150-bp sequences used to create S2 Fig.**
DrBHV, *Desmodus rotundus* betaherpesvirus.
(XLSX)

**S7 Data. Tree file for the HCMV phylogeny shown in S3 Fig.** HCMV, human cytomegalovirus.
(TRE)

## Acknowledgments

We thank Roman Biek, Barbara Marble, Mafalda Viana, and Dan Haydon for their valuable feedback on the manuscript.

## Author Contributions

**Conceptualization:** Megan E. Griffiths, Daniel G. Streicker.

**Data curation:** Alice Broos, Daniel J. Becker.

**Formal analysis:** Megan E. Griffiths.

**Funding acquisition:** Daniel G. Streicker.

**Investigation:** Megan E. Griffiths.

**Methodology:** Alice Broos, Laura M. Bergner, Diana K. Meza, Nicolas M. Suarez.

**Project administration:** Daniel G. Streicker.

**Resources:** Alice Broos, Diana K. Meza, Ana da Silva Filipe, Carlos Tello, Daniel J. Becker.

**Software:** Nicolas M. Suarez.

**Supervision:** Daniel G. Streicker.

**Visualization:** Megan E. Griffiths.

**Writing – original draft:** Megan E. Griffiths.

**Writing – review & editing:** Daniel G. Streicker.

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
