## [Editor Report · Decision Letter 0]

28 Jul 2021

Dear Dr. Griffiths, 

Thank you for submitting your manuscript entitled "Multi-strain dynamics and phylogeographic structure of a candidate viral vector supports the development of a transmissible vaccine for vampire bat rabies" for consideration as a Research Article by PLOS Biology.

Your manuscript has now been evaluated by the PLOS Biology editorial staff, as well as by an academic editor with relevant expertise, and I am writing to let you know that we would like to send your submission out for external peer review.

Please re-submit your manuscript within two working days, i.e. by Jul 30 2021 11:59PM.

Kind regards,

Paula

---

Paula Jauregui, PhD

Associate Editor

PLOS Biology

---

## [Decision Letter · Decision Letter 1]

24 Sep 2021

Dear Dr Griffiths,

Thank you for submitting your manuscript entitled "Multi-strain dynamics and phylogeographic structure of a candidate viral vector supports the development of a transmissible vaccine for vampire bat rabies" for consideration as a Research Article at PLOS Biology. Thank you also for your patience as we completed our editorial process, and please accept my apologies for the delay in providing you with our decision. Your manuscript has been evaluated by the PLOS Biology editors, an Academic Editor with relevant expertise, and by two independent reviewers.

You will see that the reviewers find your conclusions interesting and important for the field, however they also raise several concerns that would need to be addressed in order for us to consider the manuscript further for publication. Reviewer 1 mentions a general issue regarding the reliance on a small segment of glycoprotein, which will be important for you to address. Reviewer 2 would like you to clarify several points and to tone down some of the claims made in the abstract.

In light of the reviews (attached below), we will not be able to accept the current version of the manuscript, but we would welcome re-submission of a revised version that takes into account the reviewers' comments. We cannot make any decision about publication until we have seen the revised manuscript and your response to the reviewers' comments. Your revised manuscript is also likely to be sent for further evaluation by the reviewers.

We expect to receive your revised manuscript within 3 months. 

**IMPORTANT - SUBMITTING YOUR REVISION**

3. Resubmission Checklist

a) *Published Peer Review*

b) *PLOS Data Policy*

d) *Blurb*

Please also provide a blurb which (if accepted) will be included in our weekly and monthly Electronic Table of Contents, sent out to readers of PLOS Biology, and may be used to promote your article in social media. The blurb should be about 30-40 words long and is subject to editorial changes. It should, without exaggeration, entice people to read your manuscript. It should not be redundant with the title and should not contain acronyms or abbreviations. For examples, view our author guidelines: https://journals.plos.org/plosbiology/s/revising-your-manuscript#loc-blurb

Sincerely,

Ines

--

Ines Alvarez-Garcia, PhD

Senior Editor

PLOS Biology

on behalf of

Paula Jauregui, PhD

Associate Editor

PLOS Biology

Reviewers' comments

Rev. 1:

In this paper, the authors extend their previous work investigating the suitability of Desmodus rotundus betaherpesvirus (DrBHV) as a vector for a transmissible vaccine targeting rabies in vampire bats. The primary advance of this work over previous results reported by this group (Griffiths et al. 2020) is the substantially increased scope of viral sequencing and the analysis of viral sequences collected from bats that were captured and later recaptured. The increased scope of viral sequencing allows the prevalence of individual strains to be established and the analysis of recaptured animals allows the temporal dynamics of viral infections and super infections to be studied. These are fundamental properties that must be understood in the development of any transmissible vaccine platform. The authors' analyses suggest that individual strains of DrBHV reach locally high prevalence and that superinfection is possible. These are both promising and important results. Although I am generally convinced by the authors' overall conclusions, I do have some questions and concerns about methods the authors use and the robustness of some of the specific claims made throughout. Below, I detail these questions and concerns. 

The most obvious limitation of this work is the reliance on a 150 bp segment of a viral glycoprotein. This is a very small proportion of the viral genome that likely experiences strong (and potentially spatially and temporally variable) selection imposed by the host. Because of this, I was not convinced that: A) analyses of Fst for this region tell us much of anything about patterns of viral gene flow among regions, or that B) nucleotide diversity tells us much of anything about the age of a strain. In addition to violations of neutrality, the potential for recombination of this small region onto different viral genomic backgrounds also muddies the water with respect to these conclusions. I would like to see these issues addressed and their potential consequences for the authors' conclusions discussed. For instance, if nucleotide diversity has been shaped by repeated selective sweeps within strains, how would this influence the authors' conclusions? 

Figure 1b is a bit challenging to interpret quickly because colors indicate different things in the different panels. Explaining that the different panels in B are each showing one of the strains would help speed interpretation. 

Lines 144-147. I appreciate what the authors are trying to do here, but using pairwse Fst as an estimate of viral gene flow is tenuous at best because: 1) Fst depends on population size as well as gene flow and the authors estimates of strain specific prevalence suggest substantial differences in population size, 2) these results are gleaned from a 150bp fragment and so may not be representative of the genome as a whole, and 3) the glycoprotein is likely under selection that potentially varies spatially. I need to be convinced by the authors that this is a robust conclusion, because I currently am not.

Lines 154-157. Does this really show host specificity? What about just general movement corridors used by multiple bat species? This seems consistent with host specificity but not demonstrative of host specificity. 

Lines 162-174. "strain age (approximated as relative nucleotide diversity given that the nucleotide diversity of a strain would be expected to increase over time)." This seems extremely speculative and tenuous at best. Nucleotide diversity could be proportional to the age of a strain if evolution is purely neutral. But, even then, does the increase in nucleotide diversity within a strain not also depend on population size? Given the authors demonstration that strain prevalence (and presumably thus population size) varies wildly, is there actually any reason to expect that nucleotide diversity is indicative of age? This would not be true at equilibrium where we know the expected value of nucleotide diversity is equal to 4 * N * mu. Add onto this the very likely existence of selection and my confidence in this analysis becomes quite low. Here, too, I need to be convinced by the authors that this is a robust conclusion, because I currently am not.

Lines 221-226. I do not follow the logic here. The two sentences seem directly contradictory. Specifically, how is this statement "bats with higher initial strain richness acquired fewer additional strains at later time points" consistent with this statement "with strain acquisition dependent on the number of locally available strains, rather than how many an individual is infected with." I must be missing something, but no matter how many times I read these they seem to directly contradict one another.

Lines 260-264. "A major concern with using naturally occurring, innocuous, and host-specific viruses as transmissible vaccine vectors has been that cross-immunity from natural circulation of wild-type ancestors could impede vaccine spread [3]." This is, of course, only part of the challenge. If a TV rapidly dumps its antigenic cargo then it must compete with a genetically identical strain that presumably has the advantage of no longer carrying the antigenic insert. This is obviously not something that can be addressed with a field study, but this should be clearly articulated here to make it clear we have much more to learn about the dynamics of superinfection with genetically identical strains. 

Lines 437-439. "Tree prior Coalescent: constant size, and strict clock were used, 1x107 generations run with a sampling frequency of 1000, and a burn-in of 10%." This is way outside my area of expertise, but what are the consequences of assuming a constant population size and clocklike evolution in a case where: 1) the authors argue lineages arise and expand (population size not constant) and 2) the GPC is a strong candidate for periodic selection and decidedly non-clocklike evolution?

In summary, the core results of this work are robust and of broad interest. To the best of my knowledge, this is the most comprehensive study to date of the natural ecology and evolution of a betaherpesvirus that may prove to be a useful vector for a recombinant vector transmissible vaccine. The most important and novel results are the description of strain specific prevalence over space and the demonstration that superinfection appears to occur in individuals that have been studied over time. A weakness of this work, however, is that the authors over-interpret their results with respect to their more elaborate conclusions. I think the work would benefit from focusing on the robustly supported core results and eliminating the speculative conclusions about gene flow, strain dynamics over space and time, and boosting. This is not to say I believe those results are incorrect, just that the results presented here do not provide sufficiently robust support for these conclusions.

Rev. 2:

Griffiths et al. have submitted the manuscript "Multi-strain dynamics and phylogeographic structure of a candidate viral vector supports the development of a transmissible vaccine for vampire bat rabies", which attempts to unravel the phylogeography, host-use and multi-strain dynamics of the Desmodus rotundus betaherpesvirus (DrBHV) in Peru. The authors overarching aim is to lay necessary groundwork for the potential suitability of DrBHV as a transmissible vaccine (a vaccine vectored by a naturally occurring, avirulent virus) for rabies.

In summary, the authors convincingly demonstrate that DrBHV is genetically heterogeneous, with at least 11 strains (of varying intra-strain heterogeneity) being found to circulate in bat populations across Peru. Strains were found to largely vary according to host population structure as well as elevation. In addition to a considerable prevalence of DrBHV, bats were investigated for multi-strain dynamics, revealing that multiple strains of the virus are often supported by hosts. A small number of bats were also examined chronologically, revealing that infections apparently become cleared or are (re)acquired over the course of a bat's lifetime. 

However, the study does not fully evaluate whether the DrBHV system can be effectively exploited as a transmissible vaccine, as several important factors were not directly tested. The authors highlight some of these outstanding gaps in the discussion, which are very much appreciated, but mention of other issues in the discussion would be worthwhile. For example, is there experimental evidence showing that DrBHV is innocuous in bats? (This may be known, but reference to the relevant studies could be given). Can inserts be purged by selection? This could seem to be a potential issue with transmissible vaccines generally. If so, what steps could be taken to prevent insert loss in the DrBHV system? For readers unfamiliar with the topic, consideration of such issues would be useful. The abstract could also benefit from being toned down a bit, e.g. this sentence seems slightly overreaching "Our results catalyse the development of a transmissible, multi-year, self-boosting vaccine targeting the most important source of human and animal rabies in Latin America".

Overall I support the publication of this interesting manuscript. In addition to the above contextual improvements, there are some points that the authors should consider to help clarify the ms.

Major points

The Mantel tests (or their description) seem fairly crucial to the study, but they are slightly difficult to interpret and/or appear at odds with the written text. This may be a misunderstanding on my part, but it could benefit from a bit of clarification. The first test compares "Fst values for DrBHV and host microsatellites, grouped by the Department from which sequences originated". The p-value is close to being significant, but this is not necessarily noteworthy. However, the second mantel test compares either Fst values of DrBHV or bat microsatellites with "least cost distance model outputs". It is not very clear which of the former it is because in the results it reads as though the DrBHV Fsts are used, whereas in the methods, the second test "compare the matrices of microsatellite Fst values grouped by bat colony, with the least cost distance model outputs for travel between colonies". This should be cleared up, and the second test is also highly significant (P=1x10-4) so it is hard to reconcile it/interpret alongside the first mantel test. 

From the phylogeny in Fig. 1, it looks like there could be some "singleton" viral sequences that don't fit into the strain categories, e.g. the sequence at the very bottom of the phylogeny, sister to "S1" and a sequence between "S1" and "S2" that appears to be a singleton too. How were these treated, or is it just an issue with the figure graphic formatting?

On a similar theme, can strain divergence levels be somehow validated by comparing against existing full-genome sequences (if available, e.g. from single-strain infected bats)? I imagine this is probably not possible given current data availability, and the authors acknowledge the potential shortcoming of the strain ID in the discussion already, so this may be a moot point.

The precise (lat,long) sampling sites are also not given. This would be useful information for meta-analyses etc. Ditto regarding availability of qPCR data.

Figure 2 analyses of strain age, range and prevalence are also very important to the study, and overall I find the correlations to be convincing. But what is the situation at a closer scale? From the methods (although this is also a little bit unclear), it reads as though there could be multiple sampling locations per group (within a 10km range). Was sampling carried out consistently in different "groups" and are similar patterns of strain prevalence/diversity observable on this finer sampling scale. E.g. wrt strain competition, are strain number and prevalence at individual sampling locations also uncorrelated? I mention it because transmission and competition dynamics might be more meaningful at that scale. Although I must also confess that my knowledge of vampire bat ecology is insufficient to argue this convincingly! Also I fully understand that it may not be possible to test this idea given the study's sampling design, but if it were somehow possible, it might be interesting to look at briefly. More detail about the sampling strategy (sites per group, lat/long locations, distances etc) wouldn't go amiss either.

Fig 2B also appears quite bimodal. What is happening there and can the model account for that?

I am somewhat agnostic about the inclusion of the dated HCMV phylogeny, as the age of the gB4 genotype is scarcely greater than some of the others, e.g. gB3. On the other hand I can also see the reasoning behind the analysis.

Minor points

Line 106 typo "superinfection and the of"

Line 108 typo "vampire bats Peru"

Line 629. Oddly formatted reference 40

Fig 1. A very minor point but the lineages and the geographic locations vary in (very broadly) opposite directions: i.e. uppermost lineages have a more southerly distribution and vice versa. Would be visually nice to reverse that somehow.

---

## [Decision Letter · Decision Letter 2]

7 Feb 2022

Dear Dr Griffiths,

Thank you for submitting your revised Research Article entitled "Multi-strain dynamics and phylogeographic structure of a candidate viral vector supports the development of a transmissible vaccine for vampire bat rabies" for publication in PLOS Biology. I've taken over the handing of your manuscript while my colleague Dr Paula Jauregui is on maternity leave. I have now obtained advice from one of the original reviewers and have discussed their comments with the Academic Editor. 

Based on the reviews, we will probably accept this manuscript for publication, provided you satisfactorily address the remaining points raised by the reviewer. Please also make sure to address the following data and other policy-related requests.

IMPORTANT:

a) Please change your title to something more appealing and explicit. We suggest "Field genomics informs candidate vector selection and deployment strategies of a transmissible vaccine for vampire bat rabies"

b) Please address the remaining requests from reviewer #1.

c) Please address my Data Policy requests below; specifically, we need you to supply the numerical values underlying 1AB, 2ABC, 3ABCD, 4ABCD, S2, S3 (including treefiles). Please also cite the location of the data clearly in each relevant main and supplementary Fig legend, e.g. “Data underlying this Figure can be found in S1 Data”.

We expect to receive your revised manuscript within two weeks. 

*Published Peer Review History*

*Early Version*

Sincerely,

Roli Roberts

Senior Editor,

rroberts@plos.org,

PLOS Biology

DATA POLICY:

I note that you supply some raw data in your Figshare deposition; however, we also require the numerical values that underlie the figures and results of your paper be made available in one of the following forms:

Regardless of the method selected, please ensure that you provide the individual numerical values that underlie the summary data displayed in the following figure panels as they are essential for readers to assess your analysis and to reproduce it: 1AB, 2ABC, 3ABCD, 4ABCD, S2, S3 (including treefiles). NOTE: the numerical data provided should include all replicates AND the way in which the plotted mean and errors were derived (it should not present only the mean/average values).

DATA NOT SHOWN?

REVIEWER'S COMMENTS:

Reviewer #1:

The authors have done a very comprehensive job of revising their manuscript and have directly addressed my previous concerns. Including the additional analyses based on pooled full length sequences represents a significant investment by the authors in a thoughtful revision. We still disagree here and there on the strength of some specific conclusions, but this is now more a matter of opinion than anything else. There is just one last substantive issue that had not occurred to me during my original reading but which I can now not quite put to rest. This is the consequences of using affinity propagation to define strains as genetically similar clusters. What is bothering me about this is that it seems it would place a cap on the possible amount of nucleotide diversity that could be found within a "strain" before it is split by the algorithm into a new strain. Thus, only genetically similar sequences could be widespread and prevalent without being split out into separate strains. Could this lead to spurious conclusions about relationships between prevalence, range, and genetic diversity? I don't know the answer here and could be way off base, but it might be worth thinking a bit about what the null relationship should be between nucleotide diversity and prevalence or range when strains are defined as genetically similar clusters. I suspect the correct null is not the absence of a relationship.

---

## [Editor Report · Decision Letter 3]

21 Feb 2022

Dear Dr Griffiths,

On behalf of my colleagues and the Academic Editor, Andrew Read, I'm pleased to say that we can in principle accept your Research Article "Longitudinal deep sequencing informs vector selection and future deployment strategies for transmissible vaccines" for publication in PLOS Biology, provided you address any remaining formatting and reporting issues. These will be detailed in an email that will follow this letter and that you will usually receive within 2-3 business days, during which time no action is required from you. Please note that we will not be able to formally accept your manuscript and schedule it for publication until you have any requested changes.

PRESS: We frequently collaborate with press offices. If your institution or institutions have a press office, please notify them about your upcoming paper at this point, to enable them to help maximise its impact. If the press office is planning to promote your findings, we would be grateful if they could coordinate with biologypress@plos.org. If you have not yet opted out of the early version process, we ask that you notify us immediately of any press plans so that we may do so on your behalf.

Sincerely, 

Roli Roberts

Roland G Roberts, PhD 

Senior Editor 

PLOS Biology

rroberts@plos.org